# Biocompatible PVTF Coatings on Ti with Improved Bonding Strength

Weiming Lin [1], Xuzhao He [1], Xiaowei Guo [2,*], Dengfeng Xu [3] and Kui Cheng [1,*]

[1]  Center of Rehabilitation Biomedical Materials, State Key Laboratory of Silicon Materials, School of Materials Science and Engineering, Zhejiang University, Hangzhou 310027, China; 11926077@zju.edu.cn (W.L.); hf@zju.edu.cn (X.H.)

[2]  Key Laboratory of Acoustics and Vibration Precision Measuring Technology for State Market Regulation of Zhejiang Province, Zhejiang Institute of Metrology, Hangzhou 310018, China

[3]  Zhejiang Construction Engineering Group Co., Ltd., Hangzhou 310013, China; xu_deng_feng@163.com

*  Correspondence: gxw229@126.com (X.G.); chengkui@zju.edu.cn (K.C.)

**Abstract:** In this work, a poly(vinylidenefluoride-co-trifluoroethylene) (PVTF) coating on a titanium (Ti) substrate was prepared, and Ti metal surfaces were treated by physical or chemical methods to achieve a high bonding strength with PVTF. Scanning electron microscopy (SEM), atomic force microscope (AFM), and static water contact angles (WCA) were used to characterize the Ti metal surfaces. Further, mechanical stretching testing was employed to measure the bonding strength of PVTF coatings. The possible mechanism for the improved bonding strength could be the higher OH concentrations on Ti metal surfaces, which could lead to the formation of chemical bonds with the F atom of PVTF chains. Finally, a CCK-8 analysis of bone marrow mesenchymal stem cells (BMSCs) cultured on the PVTF coatings confirmed that the physical and chemical treatments had no significant differences in biocompatibility. Such a PVTF coating on a Ti substrate showed the potential of biomedical metal implants.

**Keywords:** PVTF coating; bonding strength; OH concentration; biocompatibility





## 1. Introduction

Biomedical metals have been widely used in the clinic for hundreds of years, due to their great intensity, easy matching, and favorable biocompatibility [1–3]. In orthopedics, biomedical metals are made into various shapes for bone tissue repair and implants, including guided bone regeneration membranes, artificial joints, and cranium meshes [4–6]. With the development of clinical medicine, there is an urgent need to introduce new functions to biomedical metals, e.g., bioactivity [7], antisepsis [8], and bone inducibility [9]. To achieve these bio-functions on biomedical metals, surface modification becomes one of the most important methods, especially modified bio-functional polymer on biomedical metals.

Poly(vinylidenefluoride-co-trifluoroethylene) (PVTF), such a bio-functional polymer with an excellent mechanical strength, biocompatibility and ferroelectricity [10–12], was extensively used for bone tissue repair. Luo et al. [13] reported a simple synthesis of PVTF-coated PMMA Janus membranes for guided bone regeneration. In this study, PVTF coating to promote BMSCs growth behavior, was labelled as Janus-A, and PMMA coating to inhibit the fiber-tissue growth was labelled as Janus-B. Jia et al. [14] prepared the Terfenol-D/PVTF composite polymer-modified titanium substrate with negative and positive surface potential. Compared to the positive surface potential, the negative surface potential could better promote the osteogenic differentiation of BMSCs. Tang et al. [15] prepared PVTF films on a titanium substrate with a wide range of surface potential, which was used to investigate the interfacial interactions of the cell-charged surface. It was found that optimal osteogenic differentiation of MC3T3-E1 cells was achieved on the 391 mV PVTF film.

However, rather poor interfacial adhesion and bonding exists between metal and polymer, owing to the inert surface of the polymer or metal. Hence, many methods for enhancing the metal–polymer interaction have been reported, e.g., the introduction of coupling agent, adhesive agent, plasm treatment, sandblasting, etc. For example, Jang et al. [16] investigated using modified aminosilanes as a coupling agent to promote the interfacial adhesion between aluminum metal and polyamide. Moreover, Gene and Molitor [17,18] applied adhesive agents as intermediate layers to improve the bonding of titanium alloy and fiberglass. Bhowmik et al. [19] studied surface-modified titanium sheets by plasma treatment, which could be bonded with polyimide adhesive. The results showed that the optimized time of the plasma treatment led to the maximum adhesive bond strength of the titanium. Additionally, Chai et al. [20] investigated the influence of the metal composite interface on the mechanical behaviors of Ti/CFRP/Ti laminates. It was discovered that the strength of the metal composite interface was greatly improved when the metal surfaces were sandblasted, helping to maintain the integrity of the fiber–metal laminates. Therefore, research on the improvement of metal–polymer interactions is very important.

Further, it is well known that the interaction between a metal and a polymer is generally explained by micromechanical interlocking and chemical bonds at the interface [21]. Micromechanical interlocking is essential for the spreading of a polymer on a metal with surface texturing, which contributes to the improvement of metal–polymer adhesion [22]. For example, Kim et al. [23] fabricated micro-patterns on metal surfaces as a designed surface topography in order to explain the effect of mechanical interlock on adhesion strength. The results found that the molecular dissipation of the polymer in the vicinity of the interface was the primary cause of the practical separation energy, and the loading mode controlled the mechanical interlock effect. In addition, Quintana et al. [24] studied surface modification pretreatments on laser direct joining of fiberglass-reinforced polyamide to steel, and found a tight dependence of mechanical interlock with micro-patterns on the joint's failure force. Moreover, chemical bonds also played important roles in bonding strength. Chen et al. [25] revealed the bonding mechanism of a laminated Cr-coated steel-ethylene acrylic acid strip. The result showed that hydrogen bonds and covalent bonds to the interface were explored by X-ray photoelectron spectroscopy for the first time. Yusuke et al. [26] discussed the chemical interaction between aluminum and polyamide 6, and characterized the joint interface with AFM-IR. It was found that –CONH in PA6 formed the hydrogen bond with the hydroxyl groups on the aluminum surface, which made a contribution to the bonding strength.

Although many studies have been undertaken to improve the metal–polymer interaction, new materials, such as coupling and adhesive agents, might have impacts on the biocompatibility of biopolymer-modified metals. Hence, this study focused on a method that could enhance the metal–polymer interaction while not affecting biocompatibility. In this work, the PVTF coatings on Ti substrates modified by physical and chemical treatments were prepared. The bonding strength of PVTF coatings with different treatments was investigated. The probable mechanism of improved bonding strength of PVTF coatings was discussed. The biocompatibility of various PVTF coatings was also studied.

## 2. Materials and Methods

### 2.1. Materials

PVTF (70/30) powder was purchased from Piezotech (Pierre-Bénite, France). N, N-dimethylformamide (DMF, 99.5%), hydrochloric acid (HCl, 36%), nitric acid (HNO$_3$, 68%), hydrofluoric acid (HF, 40%), acetone (99.5%), and absolute ethanol (99.7%) were supplied from Sinopharm Chemical Reagent Co., Ltd. (Shanghai, China). 400-, 2000- and 5000-grit sandpapers were bought from Xinge Abrasive (Suzhou, China). Ti metal with thickness of 100 μm was obtained from Boti Titanium Products Co., Ltd. (Baoji, China).

### 2.2. Ti Metal Surfaces with Physical and Chemical Treatments

Ti metal surfaces were polished by 400-, 2000-, and 5000-grit sandpapers, soaked in 36% HCl, 10% HF (40% HF: $H_2O$ = 1:3 in volume), and mixed acid consisting of HF, $HNO_3$, and $H_2O$ (1:2:4.2 in volume) solution for 1 min, respectively. Then, the treated Ti metals were cleaned in acetone, deionized water and absolute ethanol in ultrasonic bath for 10 min, respectively.

### 2.3. Preparation of PVTF Coatings and Specimens

The PVTF coatings on the Ti substrate were prepared by following the method [27]. Briefly, 18 wt% PVTF powder was dissolved in DMF and stirred magnetically for 2 h to obtain the completely dissolved solution. The PVTF solution was cast on Ti substrate, then prepared by the annealing method, finally forming the crystallization of PVTF at 210 °C for 1 h. The cyanoacrylate adhesive, which was the liquid condition in native, was used to coat one of the sides of mold and the sample. After the solvent in cyanoacrylate adhesive evaporation, one of the sides of mold and the sample was preliminarily bonded. The other sides of the mold were coated with the sample following the above-mentioned method. The cyanoacrylate adhesive formed the adhesive film, and the specimens were made successfully overnight. As shown in Figure 1, the specimens were made for mechanical stretching tests.

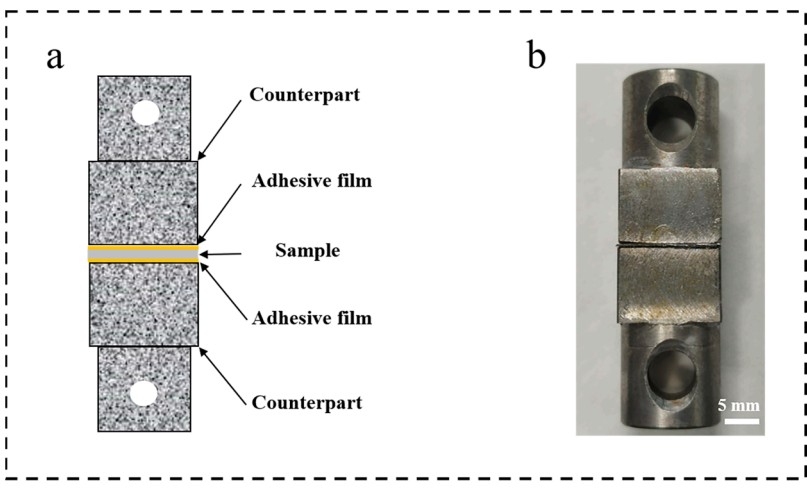

**Figure 1.** The specimen after bonding. The mold of the (**a**) schematic diagram and (**b**) object.

### 2.4. Characterization of PVTF Coatings and Ti Metal Surfaces

The morphology of PVTF coatings and Ti metal surfaces modified with physical and chemical treatments was observed by scanning electron microscopy (SEM; Hitachi, Tokyo, Japan, SU-70, 3.0 kV) with an 8.0 mm depth of field. Fourier transform infrared spectroscopy (FTIR) of PVTF coatings was recorded by an infrared spectrometer (Nicolet is50 Thermo Fisher, Waltham, MA, USA) with a range of 700 $cm^{-1}$~1500 $cm^{-1}$. X-ray diffraction patterns (XRD, X-pert Powder, PANalytical, Malvern, UK, Cu Kα, 40 kV) were used to determine the crystalline phase of PVTF coatings, where the range was from 10 to 40°, and with the scan rate of 2° $min^{-1}$. The crystalline content was evaluated with a differential scanning calorimeter (DSC, PE DSC 7, Perkin Elmer, Waltham, MA, USA) based on melting enthalpy. To eliminate the thermal history of the samples, the samples were heated from 50 °C to 200 °C at a heating rate of 10 °C/min. The thermal stability of the samples was determined using an STA analyzer (TGA, Mettler Toledo, Greifensee, Switzerland) in the temperature range of 50–600 °C, with a heating rate of 10 °C/min. The roughness, which was calculated in terms of "Sa", and topography of Ti metal surfaces were characterized by atomic force microscope (AFM, NTEGRA Spectra C, NTMDT, HQ: NSC18/Pt) force–distance curves with "contact" mode for three replicates. The static water contact angles (WCA) of the surfaces were observed with a contact angle

meter (Dataphysics, OCA20, Filderstadt, Germany) and measured by three replicates. The bonding strength of samples was measured by a material testing machine (Zwick/Roell Z020, Ulm, Germany) at a speed of 1 mm/min at room temperature for four replicates. To analyze the concentration of OH groups and show the change in the difference between $TiOH_T$ and $TiOH_B$, Casa X-ray photoelectron spectrometer (CasaXPS, AXIS SUPRA, Krato, Manchester, England, Al K$\alpha$) from 525 eV to 540 eV was used to obtain the oxygen elements on Ti metal surface.

### 2.5. Cell Culture

The Laboratory Animal Welfare and Ethics Committee of Zhejiang University approves all experiments on animals. Bone marrow mesenchymal stem cells (BMSCs) were extracted from three-week-old male Sprague Dawley (SD) rats (Ethics Code: ZJU20220279) and cultured in minimum essential medium (MEM Alpha, Gibco, Grand Island, NY, USA), which contains 10% fetal bovine serum (Cellmax, Beijing, China), 1% antibiotic solution containing $1 \times 10^4$ units $mL^{-1}$ penicillin, 1% sodium pyruvate, $1 \times 10^4$ $\mu g$ $mL^{-1}$ streptomycin, and 1% MEM nonessential amino acids (all from Gibco, Grand Island, NY, USA). BMSCs were cultured in a humid atmosphere of 5.0% $CO_2$ at 37 °C. BMSCs on the samples were trypsinized with 0.25% trypsin containing 1 mM ethylenediamine triacetic acid (Gibco).

### 2.6. Cell Vitality Assays

A 500 $\mu L$ suspension of BMSCs, with a density of $5 \times 10^4$/well, was seeded on the PVTF coatings with the size of $1 \times 1$ $cm^2$, and cultured for 1 day and 3 days. The cell counting kit-8 (CCK-8, Dojindo Laboratories) assay was used to evaluate cell ability. In brief, the samples were transferred to new 24-well plates, and washed three times in PBS to remove the residual medium. A total of 550 $\mu L$ of the new medium containing 10 vol% CCK-8 solution was added and incubated for 2 h. After that, 120 $\mu L$ of the new medium was added to the 96-well and set for four replicates, then measured by a microplate reader (MULTISKAN MK3, Thermo Fisher, Waltham, MA, USA) at the absorbance of 450 nm.

### 2.7. Statistical Analysis

A one-way ANOVA with Tukey's post hoc test was used to perform the statistical analysis. A value of $p < 0.05$ was considered statistically significant (* $p < 0.05$, ** $p < 0.01$, *** $p < 0.001$). All data were represented as mean $\pm$ standard deviation (S.D.).

## 3. Results and Discussion

### 3.1. Characterization of PVTF Coatings

SEM images of PVTF coatings showed that typical liner grains appeared due to the annealing process (Figure 2a). After that, PVTF coating formed the crystallization of the β-phase, which was closely related to piezoelectricity [28]. As shown in Figure 2b, SEM images of the cross-section showed the thickness, which was about 130 $\mu m$, and interfaces of PVTF coated on Ti metal. The high magnification side view SEM images of Ti/PVTF interface confirmed that PVTF was closely adhered to Ti metal (Figure 2c).

The FTIR spectra of pure and all PVTF coatings (Figure 3a) showed absorption bands at 840 $cm^{-1}$, 1285 $cm^{-1}$, and 1400 $cm^{-1}$, which were indexed to β-phase [29]. In addition, the peak at 880 $cm^{-1}$ and from 1072 $cm^{-1}$ to 1176 $cm^{-1}$ was associated with –$CF_2$ bond in-plane rocking deformation and the α, β, and γ phases, respectively. As shown in Figure 3b, XRD patterns also confirmed that pure and all PVTF coated on Ti samples formed the β-phase at the peaks of 2θ = 19.6° [30]. The DSC of pure and all PVTF coated on Ti samples both showed two distinct peaks, one peak at 80–105 °C and another peak at 155 °C. The samples had almost the same transition temperature (Tc) and melting temperature (Tm), indicating that the crystallinity of pure and all PVTF coated on Ti samples was similar (Figure 3c). The TGA thermographs were shown in Figure 3d. All samples showed a major degradation in the temperature region of 330–550 °C. The weight loss of pure and all PVTF coated on Ti samples was 95% and about 25%, respectively.

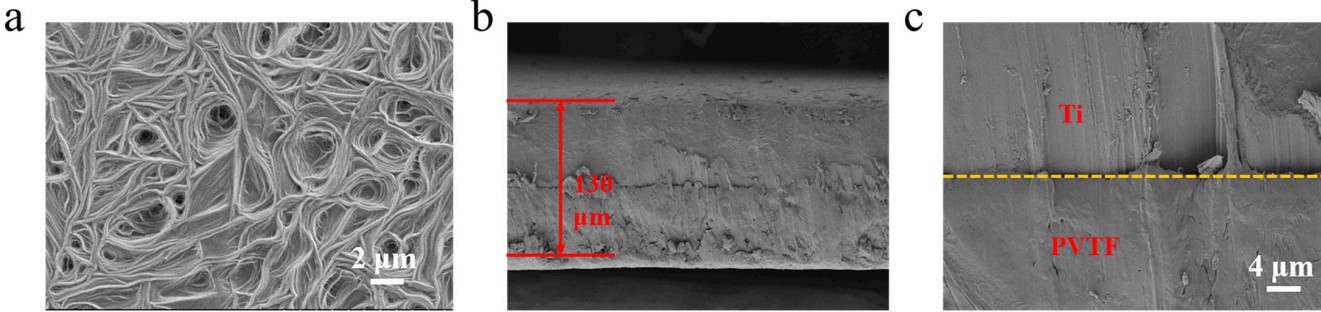

**Figure 2.** Characterization of PVTF coatings. The SEM of (**a**) PVTF, and (**b**,**c**) cross-section of PVTF coated on Ti metal. (The dashed line represented the interface of Ti metal and PVTF).

**Figure 3.** The (**a**) FTIR spectra, (**b**) XRD patterns, (**c**) DSC, and (**d**) TGA thermographs of pure and all PVTF coated on Ti samples.

### 3.2. Characterization of Ti Metal Surfaces after Physical and Chemical Treatments

To enhance the bonding strength of PVTF coatings, physical and chemical treatments were made for Ti metal surfaces. SEM images of Ti metal surfaces modified by physical treatment showed that Ti metal surfaces became increasingly smooth as the mesh number of sandpaper was increased (Figure 4a–c). Moreover, the surface of topography and roughness of these Ti metal surfaces were also revealed in AFM images (Figure 5a–c). AFM images showed that Ti metal surfaces with 400-grit, 2000-grit and 5000-grit sandpaper treatments had a series of gullies. With the number of sandpapers, the gullies on Ti metal surfaces became shallow. As shown in Table 1, the roughness of Ti metals presented the tendency to decrease, and the roughness of samples with 400-grit, 2000-grit and 5000-grit sandpaper treatments were 557.4 ± 75.4 nm, 320.9 ± 18.2 nm and 216.3 ± 76.1 nm, respectively. As shown in Figure 4d–f, the morphology of Ti metal surfaces was significantly diverse with chemical treatments. AFM images showed that a Ti metal surface treated by HCl solution had small holes, a Ti metal surface treated by HF solution formed many irregular pits, and a Ti metal surface treated by mixed acid solution formed regular pits (Figure 5d–f). After treatment with HCl, HF, and mixed acid solutions, the roughness of the Ti metal surfaces were 440.3 ± 51.6 nm, 691.9 ± 41.2 nm, and 464.2 ± 54.9 nm, respectively. In previous studies [31,32], it was reported that the roughness was related to the hydrophilicity/hydrophobicity. For this reason, the static water contact angle (WCA) of Ti metal surfaces was also measured. As shown in Table 2, the WCA of Ti metal surfaces treated with 400-, 2000- and 5000-grit sandpaper were 92.3 ± 3.1°, 80.5 ± 2.9°, and 78.9 ± 0.9°, respectively. The results were basically consistent with the roughness, indicating that the hydrophilicity increased with the decrease of roughness obtained by the physical method. In addition, the WCAs of the Ti metal surfaces were 54.7 ± 2.7°, 15.0 ± 2.7°, and 9.9 ± 1.9° after treatment with HCl, HF and mixed acid solutions, respectively. The results showed an inconsistent trend compared to the above-mentioned results, which might be due to the fact the hydrophilicity/hydrophobicity obtained by the chemical method was influenced by other factors, e.g., chemical bonds [33,34]. It was worth noting that the relationship between roughness and WCA of Ti metal surfaces with different physical and chemical treatments was exhibited in Figure S1, Supplementary File.

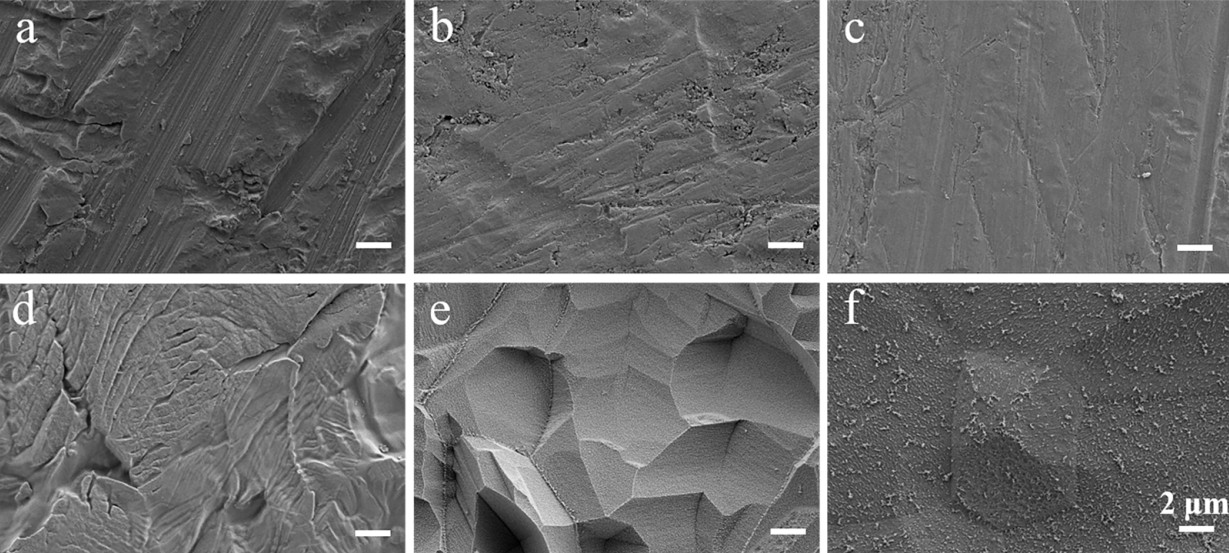

**Figure 4.** SEM images of Ti metal surfaces after physical and chemical treatments. Ti metal surfaces were treated by (**a**) 400-, (**b**) 2000-, (**c**) 5000-grit sandpaper, (**d**) 36% HCl, (**e**) 10% HF and (**f**) mixed acid solutions for 1 min, respectively.

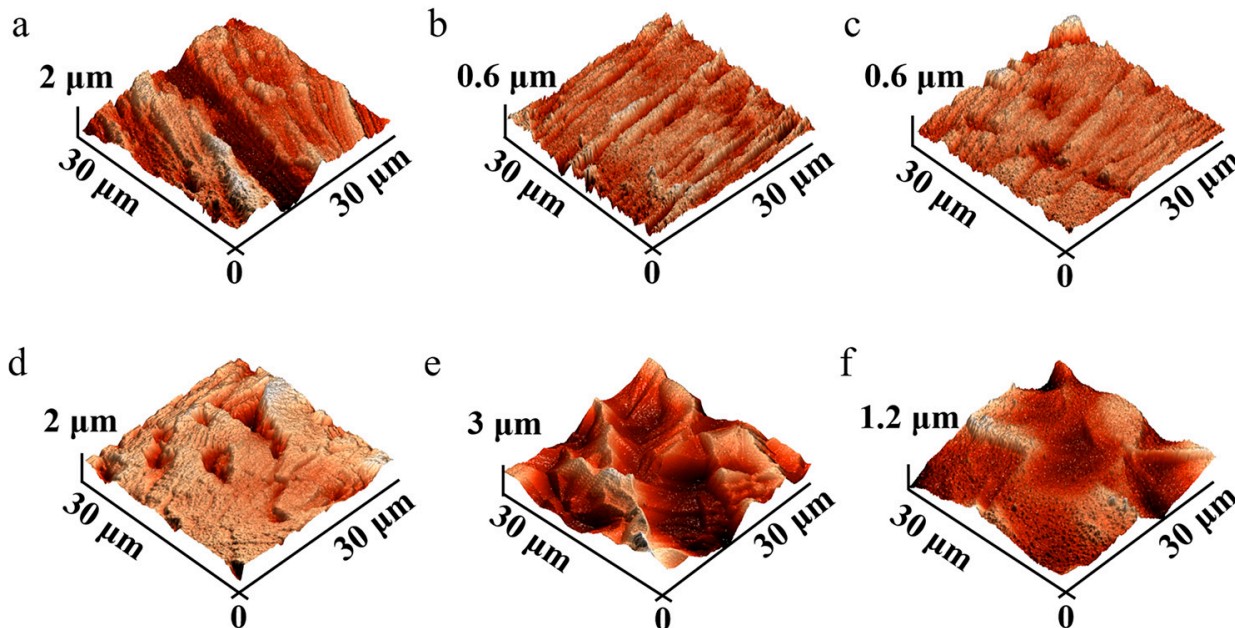

**Figure 5.** AFM images of Ti metal surfaces after physical and chemical treatments. Ti metal surfaces were treated by (**a**) 400-, (**b**) 2000-, (**c**) 5000-grit sandpaper, (**d**) 36% HCl, (**e**) 10% HF and (**f**) mixed acid solutions for 1 min, respectively.

**Table 1.** The roughness of samples with different treatments were measured by AFM.

| Treatment | Roughness (nm) |
|---|---|
| 400-grit | 557.4 ± 75.4 |
| 2000-grit | 320.9 ± 18.2 |
| 5000-grit | 216.3 ± 76.4 |
| HCl | 440.3 ± 51.6 |
| HF | 691.9 ± 41.2 |
| Mixed acid | 464.2 ± 54.9 |

**Table 2.** The WCAs of samples with different treatments.

| Treatment | WCA (°) |
|---|---|
| 400-grit | 92.3 ± 3.1 |
| 2000-grit | 80.5 ± 2.9 |
| 5000-grit | 78.9 ± 0.9 |
| HCl | 54.7 ± 2.7 |
| HF | 15.0 ± 2.7 |
| Mixed acid | 9.9 ± 1.9 |

*3.3. The Bonding Strength of PVTF Coatings*

Ti metal surfaces modified by physical or chemical methods could obtain different roughness and topography, leading to the different bonding strengths of PVTF coatings. The bonding strength was calculated as follows [35]:

$$\sigma = F/S, \tag{1}$$

where $\sigma$, F, and S are bonding strength, the force at rupture, which was produced by the separation of specimen after stretching in a mechanical test, and the area of PVTF coatings, respectively. The specimen after bonding was used to measure the bonding strength.

### 3.3.1. The Effect of Physical Treatment on Bonding Strength of PVTF Coatings

As shown in Figure 6a, the bonding strength of samples increased with a decrease in roughness. The bonding strength of samples with 400-grit, 2000-grit and 5000-grit sandpaper treatments were 2.10 ± 0.41 MPa, 4.67 ± 0.34 MPa, and 5.06 ± 0.85 MPa, respectively, and the untreated sample was 0.68 ± 0.20 MPa, shown in Table 3. In addition, the morphology of the fracture after stretching was quite different when samples happened to break (Figure 7a–c). The morphology of samples with 400-grit sandpaper treatment after stretching showed that only peripheral PVTF was destroyed, and the center section of PVTF remained intact, indicating that the center section of PVTF was not effectively participated in bonding strength. The reason might be that the excessive roughness of Ti metal surfaces modified by sanding treatment easily formed sharp edges and corners, causing stress concentration and reducing wettability, which is not beneficial for PVTF chains to spread on the metal surface [36]. As the number of sandpaper meshes increased, the roughness of Ti metal decreased, which resulted in Ti metal surface flattening and more area of PVTF took part in bonding strength. Therefore, samples treated with 5000-grit sandpaper had the highest bonding strength in physical methods.

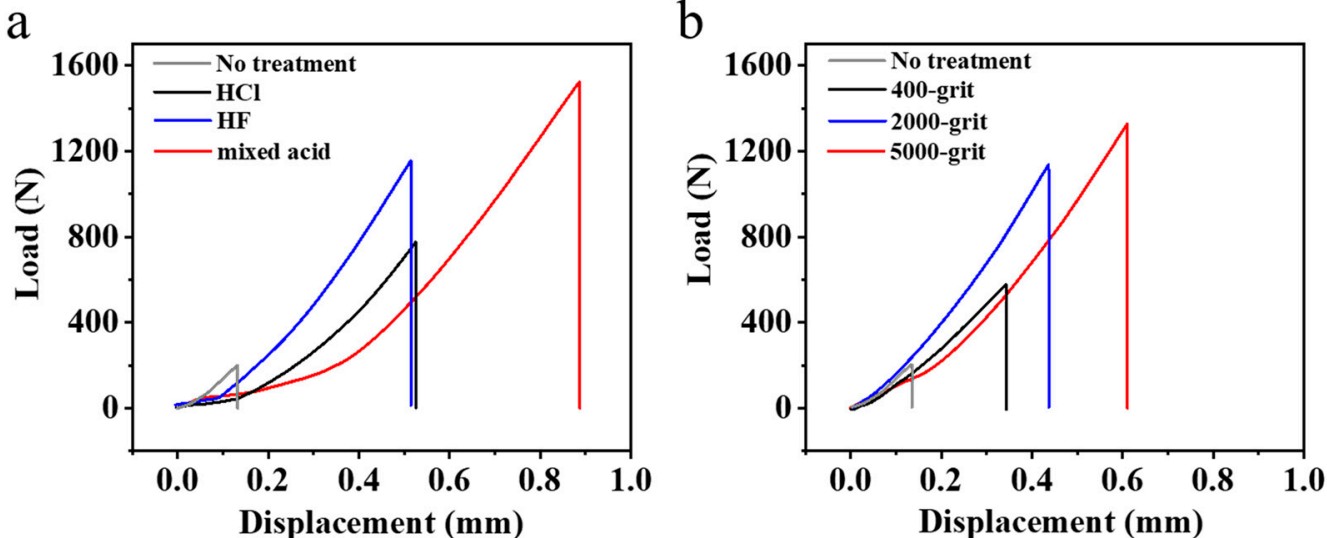

**Figure 6.** Bonding strength of samples after physical and chemical treatments. The bonding strength of samples treated by (**a**) 400-,2000-,5000-grit sandpaper, (**b**) 36% HCl, 10% HF and mixed acid solutions for 1 min, respectively, compared to the sample with no treatment.

**Table 3.** The bonding strength of samples with different treatments were measured by mechanical stretching tests.

| Treatment | Bonding Strength (MPa) |
| --- | --- |
| No treatment | 0.68 ± 0.20 |
| 400-grit | 2.10 ± 0.41 |
| 2000-grit | 4.67 ± 0.34 |
| 5000-grit | 5.06 ± 0.85 |
| HCl | 3.22 ± 0.03 |
| HF | 4.98 ± 0.35 |
| Mixed acid | 6.88 ± 0.11 |

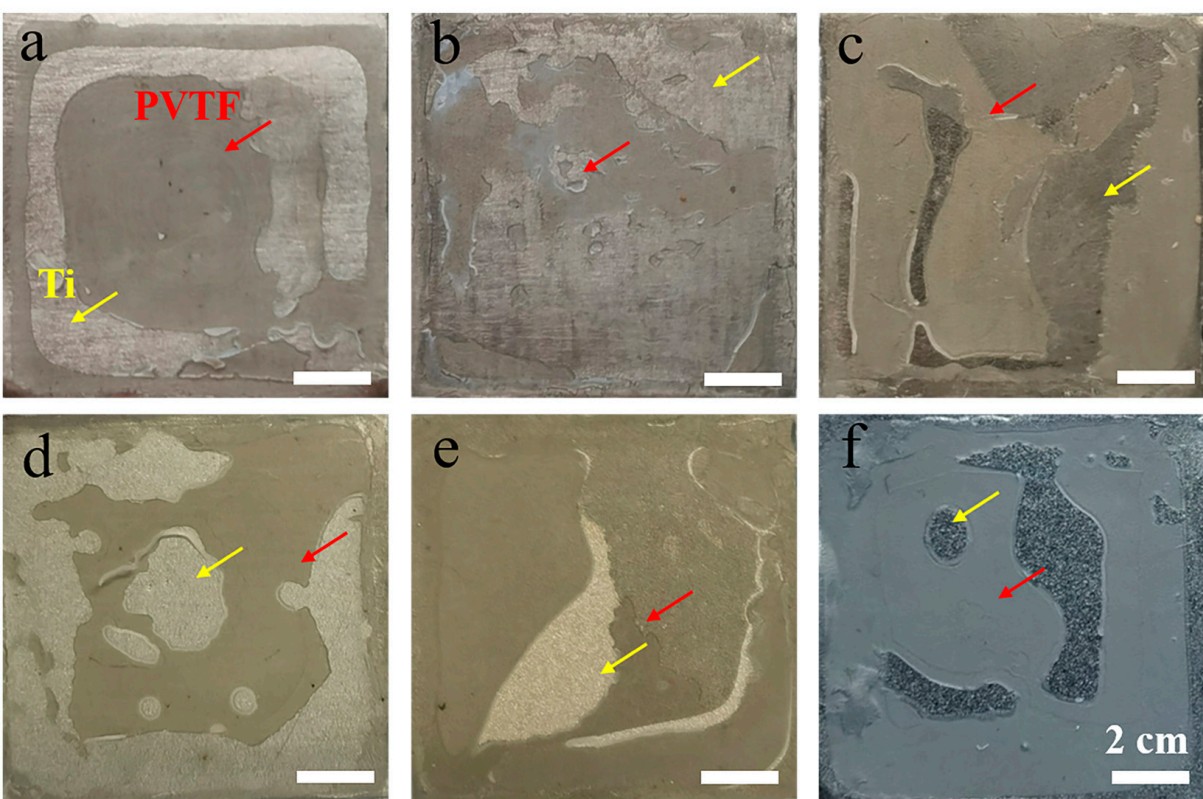

**Figure 7.** The morphology of fracture of samples after stretching. The fracture of samples treated by (**a**) 400-, (**b**) 2000-, (**c**) 5000-grit sandpaper, (**d**) 36% HCl, (**e**) 10% HF and (**f**) mixed acid solutions for 1 min, respectively. Red arrows represented PVTF, yellow arrows represented Ti metal.

### 3.3.2. The Effect of Chemical Treatment on Bonding Strength of PVTF Coatings

The bonding strength of samples with chemical treatments was shown in Figure 6b. The bonding strength of samples treated by HCl, HF, and mixed acid solutions were $3.22 \pm 0.03$ MPa, $4.98 \pm 0.35$ MPa, and $6.86 \pm 0.11$ MPa, respectively, which is shown in Table 3. The Ti metal surfaces treated with the HCl solution have fewer pits, which made it difficult for PVTF to bond with Ti metal. The morphology of the fracture after stretching showed a small amount of the PVTF participated in bonding with the Ti metal (Figure 7d). Al Hussaini et al. [37] also found that the Ti metal surface modified by the HCl solution treatment formed an oxide layer, which might explain the low bonding strength. The fracture morphology after stretching of samples with the HF solution treatment (Figure 7e) mainly manifested as cohesive destruction. A Ti metal surface with high roughness and many pits, because of the dissolution of Ti in HF solution, might be advantageous for PVTF to spread and form mechanical interlocking. Obviously, the fracture morphology after stretching of samples with mixed acid solution treatment, which had the highest bonding strength, also showed cohesive destruction (Figure 7f). The extreme hydrophilicity and formation of many anisotropic pits and particulate matter on the Ti metal surface due to the passivating effect of $HNO_3$ and the dissolution of Ti in HF might be the reason for the highest bonding strength.

### 3.4. The Possible Mechanism of Bonding Strength of PVTF Coatings

To further explore the possible mechanism of the high bonding strength of the PVTF coatings, XPS spectra were employed to detect functional groups on the Ti metal surface, especially for the OH group, which could theoretically form hydrogen bonds with the F atom of PVTF chains. As shown in Figure 8a,b, an XPS of O 1s peak, which was to analyze the OH group, was divided into four subpeaks: the peak near 529.9 eV was attributed to the

O in the surface oxide lattices (denoted by $TiO_2$), the peak near 531.4 eV was assigned to the O in the basic $TiOH_B$ (denoted by $TiOH_B$, bridging OH group), the peak near 532.2 eV was attributed to the O in the acidic $TiOH_T$ groups (denoted by $TiOH_T$, terminal OH group) [38] and the peak near 533.4 eV was assigned to the $O_2^{2-}$ groups [39]. Since different types of OH groups could result in different polarities, it was very important to calculate the area ratio of $TiOH_T/TiOH_B$, which could represent the surface property of the material [40]. The relative peak area ratios are shown in Table 4. The results showed that 5000-grit sandpaper and mixed acid group of $TiOH_T/TiOH_B$ were 1.00 and 0.43, respectively. More importantly, the total OH group concentration of mixed acid group was 42.5, which was more than 5000-grit sandpaper group, 30.02. That might explain why the bonding strength of the mixed acid group was higher.

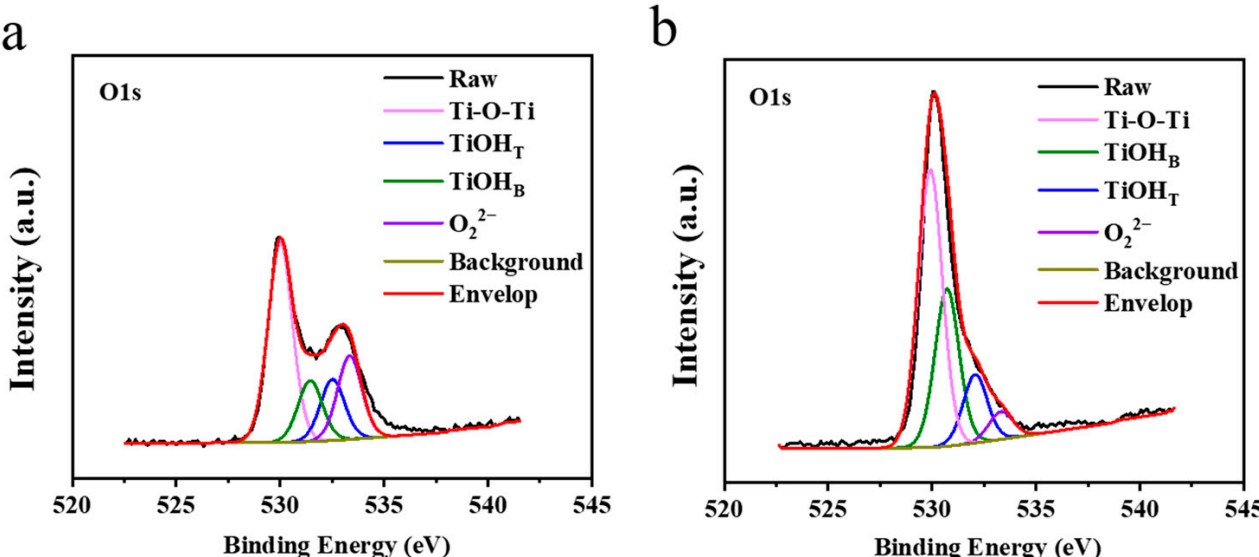

**Figure 8.** The deconvolution of XPS O1s spectra on Ti surfaces with 5000-grit sandpaper and mixed acid treatments. XPS O1s spectra of Ti surfaces with (**a**) 5000-grit sandpaper and (**b**) mixed acid treatments.

**Table 4.** Percentage areas of the $TiOH_B$, $TiOH_T$, $O_2^{2-}$, and $TiO_2$ peaks obtained by deconvoluting the XPS O1s spectra of Ti surfaces with 5000-grit sandpaper and mixed acid treatments.

|  | $TiOH_B$ | $TiOH_T$ | $O_2^{2-}$ | Ti-O-Ti | $TiOH_T/TiOH_B$ | Total OH |
|---|---|---|---|---|---|---|
| 5000-grit sandpaper | 15.01 | 15.01 | 20.58 | 49.39 | 1.00 | 30.02 |
| Mixed acid | 29.63 | 12.87 | 5.21 | 52.28 | 0.43 | 42.5 |

*3.5. The Biocompatibility of PVTF Coatings*

To understand whether the Ti metal surface being treated by physical or chemical methods had an influence on PVTF coatings, CCK-8 tests were used to evaluate the biocompatibility. With the PVTF serving as the control group, BMSCs were seeded on PVTF coatings with 5000-grit sandpaper and mixed acid solution treatment, and were cultured for 1 day and 3 days. As shown in Figure 9, compared with the control group, the OD value of PVTF coatings with 5000-grit sandpaper and mixed acid solution treatment had good biocompatibility, and showed no significant difference from each other. This indicated that physical and chemical methods had no effect on the biocompatibility of PVTF coatings.

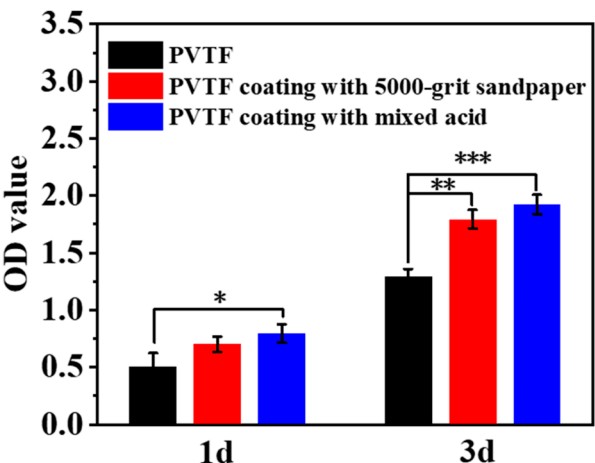

**Figure 9.** Cell viability of different PVTF coatings with 5000-grit sandpaper and mixed acid solution treatment was measured by CCK-8 assay. (* $p < 0.1$, ** $p < 0.01$, *** $p < 0.001$).

## 4. Conclusions

In this work, Ti metal surfaces were treated by physical or chemical methods, and PVTF coatings on Ti substrates were successfully prepared. The bonding strength of PVTF coatings, and a possible mechanism of such a high bonding strength, was investigated. The biocompatibility of PVTF coatings on the Ti substrate modified by 5000-grit sandpaper and mixed acid solution treatments was also examined. The conclusion points are as follows:

1.  The lowest roughness on the Ti metal surfaces with physical treatments showed the highest bonding strength of PVTF coatings. The most hydrophilicity on the Ti metal surfaces with chemical treatments showed the highest bonding strength of PVTF coatings;
2.  The total OH concentrations on the Ti metal surfaces modified by mixed acid treatment were higher than that modified by 5000-grit sandpaper treatment. This might be the underlying mechanism of the higher bonding strength;
3.  CCK-8 results indicated that PVTF coatings with physical and chemical treatments had good biocompatibility, and showed no significant difference. Biocompatible PVTF coatings on Ti with improved bonding strength exhibit broad application prospects as biomedical materials.

**Supplementary Materials:** The following supporting information can be downloaded at: https://www.mdpi.com/article/10.3390/coatings13071224/s1, Figure S1: The distribution and fitting data for the roughness and static water contact angles estimation.

**Author Contributions:** Methodology, W.L.; investigation, X.H. and X.G.; data curation, W.L. and X.G.; writing—original draft preparation, W.L.; visualization, D.X., conceptualization, K.C. All authors have read and agreed to the published version of the manuscript.

**Funding:** This research was funded by National Natural Science Foundation of China (52271252), Science and Technology Program Project of the State Administration of Market Supervision (2022MK046) and Eyas Program Incubation Project of Zhejiang Provincial Administration for Market Regulation (No. CY2022001).

**Institutional Review Board Statement:** Not applicable.

**Informed Consent Statement:** Not applicable.

**Data Availability Statement:** The authors confirm that the data supporting the findings of this study are available within the article.

**Conflicts of Interest:** The authors declare no conflict of interest.

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
