# Peer review of "Biocompatible PVTF Coatings on Ti with Improved Bonding Strength"

_coatings, doi:10.3390/coatings13071224_

Round 1

Reviewer 1 Report

The authors reported an investigation of the PVTF Coatings on Ti with Improved Bonding  Strength effect

The topic is interesting. However, experiments include significant drawbacks. It does not allow for the formulation of certain conclusions.

1.            Additional information about the method of deposition of cyanoacrylate adhesive should be supplied. The role of this agent was not studied and discussed. This is obligatory.

2.            AFM micrographs should be presented in the new Figure. All Z-scale should be rescaled to the same value. It is not clear which roughness parameters were estimated (Sa or Sq ?). A more detailed discussion about AFM results should be provided.

3.            It is obvious that roughness greatly affects the results of WCA values. Provided  WAC data does not correlate with roughness. Surface chemistry (chemical bonds) formed after chemical treatments determine WAC and that issue should be carefully studied as it is the most important for adhesion force.

4.            It is not clear how the values of adhesion strength were calculated form the data in Fig.4. Some example is needed.

5.            In Fig. 4 there is a mistake in the captions.

6.            Section 3.4 devoted to XPS measurements which is the most important for understanding the chemical forces for bonding strength is very unclear. The discussion is very cursory. First, I am surprised that the authors claim that hydrogen bonds can be responsible for adhesin force. Hydrogen bonds are very weak and based on physical interactions. It is obvious that strong chemical bonds should be responsible for increasing adhesion force. For me, it is unclear: “surface oxide lattices (denoted by TiO2), the peak near 531.4 eV was assigned to O in basic TiOHT (denoted by TiOHT, bridging OH group), the peak near 532.2 eV was attributed to the O in acidic TiOHB groups (denoted by TiOHB, terminal OH group) and the peak near 533.4 eV was assigned to the O2 2- groups [31]”.The authors should provide an appropriate Figure that shows possible molecular arrangements on the Ti surface.

I do not understand what does it mean total OH ? The authors do not provide XPS results for the rest samples. Due to this reason, the experiment is incomplete, and any serious conclusions should not be provided.

7.            Paragraph 3.5 presents only results for two samples. It is not clear what does it mean OD means. Any discussion is provided. Due to this reason, I recommend removing this paragraph since serious biocompatibility studies required much more effort. The title should be only limited to the studies about bonding Bonding  Strength effect.

8.            The manuscript includes many typos and grammar issues, e.g. many sentences begin with “And”.

The manuscript includes many typos and grammar issues, e.g. many sentences begin with “And”. The English should be revised by a native speaker.

Reviewer 2 Report

In this paper, authors investigate a method to enhance the metal-polymer interaction based on PVTF coatings on Ti substrates modified by physical and chemical treatments, trying not affecting biocompatibility. The bonding strength and biocompatibility of PVTF coatings with different treatments was analyzed.

I consider the general approach is correct, and the techniques used to characterize the samples are suitable, but very poorly described and their results not rigorously shown, therefore, a general improvement of its presentation is necessary so that the article has the required quality.

- In Materials, you use PVTF (70/30), though I suppose each number corresponds to each component; it is necessary to described in more detail referring to each component and citing relative weight percentage -or volume-.

- In Materials, I miss the purity or quality data of the organic solvents. Sometimes these data are relevant.

- In “2.4. Characterization of PVTF coatings”, it is necessary to provide methods description, in almost all of them, and also add the number of samples analyzed to give the data with each technique. For example, in AFM methodology, you should add the operational mode and the probes used, and the number of samples analyzed to provide the roughness and associated error estimation.

- In line 141, authors say “it was reported that the roughness was related to the hydrophilicity/hydrophobicity”, but you should add “ratio”.

- In “Table 1. The Roughness of samples with different treatments” you should add that this parameter was measured by AFM. You should add the units of the measurements.

- In Table 2, you should add the units of the measurements.

- In Figure 3, are all the SEM images imaged using the same scale? I mean, if all of them correspond to 2 mm line scale. The AFM images are terribly small. They should be bigger enough to see the details of the surface. I consider that these AFM images must be in a separate figure in the main manuscript, in one for themselves to be able to see in detail the surfaces of the samples, since the roughness and homogeneity of the surfaces are important data here.

- Please provide distribution and fitting data for the roughness and static water contact angles estimation at the supplementary data.

- In figure 5, please, explain better what arrows indicate in figure caption.

Please, improve the general language of the article. Some sentences are elaborated in a very poor language, too schematic. In the conclusions it would be better that instead of numbers, some ideas were better threaded with others, helping the general understanding of the article.

Reviewer 3 Report

The authors reported poly(vinylidenefluoride-co-trifluoroethylene) (PVTF) coating on titanium (Ti) metal surface treated by different physical and chemical methods

The manuscript must be improved via additional experiments and significant English language editing

Major comments

1- Thickness of PVTF coat on Ti metal should be measured in addition to High magnification side view SEM images of PVTF/Ti interfaces should be taken.

2- Surface scratch test along with SEM images after scratch of all PVTF coated Ti samples should be added 

3-Thermal analyses by TGA and DSC of pure  and all PVTF coated Ti samples should be performed.

4- XRD and FT-IR   of all PVTF coated Ti samples should be collected.

5- Imaging of cell morphologies on surfaces of pure and PVTF coated Ti samples should be performed  after 1 and 3 days of culture

English Language of the manuscript is of low quality. 

Significant English language editing in terms of writing quality, grammar correctness and clarity must be performed

Reviewer 4 Report

Review: Biocompatible PVTF Coatings on Ti with Improved Bonding Strength

This work studies the adherence of the copolymer PVTF to the surface of titanium metal as a function of the surface roughness. Different degrees of roughness were obtained by various physical or chemical treatments of the metal surface, and were characterized by SEM, AFM and contact-angle analyses. The strength of the adherence was measured quantitatively by a material testing machine in units of MPa. It was found that stronger bonding were formed on smoother surface, and the PVTF coating is biocompatible with certain living cells.

The results are interesting and applicable, but the following points should be addressed before it can be published.

1. Lines 121 and 133: Should ne “showed” instead of “observed”.

2. Lines 121-122: What are typical linear grains? Why is it due to annealing?

3. Line 124: Should be “which were”.

4. Line 140: Can it be concluded that HF treatment has no effect on the roughness?

5. Line 142: The abbreviation WCA is not defined.

6. Fig. 3: The AFM images are too small so no details or numbers can be observed.

7. Line 155: Should be “after physical or chemical treatment”.

8. Line 162: Should be “force” instead of “pull”.

9. Line 163: A sentence cannot start with the word “and”.

10. Lines 167 and 186: These values should be given in a table.

11. Fig. 5: a) I do not think that conclusions can be drawn from one set of experiments. At least four repetitions are needed. b) A quantitative analysis by image processing is required here.

Review: Biocompatible PVTF Coatings on Ti with Improved Bonding Strength

This work studies the adherence of the copolymer PVTF to the surface of titanium metal as a function of the surface roughness. Different degrees of roughness were obtained by various physical or chemical treatments of the metal surface, and were characterized by SEM, AFM and contact-angle analyses. The strength of the adherence was measured quantitatively by a material testing machine in units of MPa. It was found that stronger bonding were formed on smoother surface, and the PVTF coating is biocompatible with certain living cells.

The results are interesting and applicable, but the following points should be addressed before it can be published.

1. Lines 121 and 133: Should ne “showed” instead of “observed”.

2. Lines 121-122: What are typical linear grains? Why is it due to annealing?

3. Line 124: Should be “which were”.

4. Line 140: Can it be concluded that HF treatment has no effect on the roughness?

5. Line 142: The abbreviation WCA is not defined.

6. Fig. 3: The AFM images are too small so no details or numbers can be observed.

7. Line 155: Should be “after physical or chemical treatment”.

8. Line 162: Should be “force” instead of “pull”.

9. Line 163: A sentence cannot start with the word “and”.

10. Lines 167 and 186: These values should be given in a table.

11. Fig. 5: a) I do not think that conclusions can be drawn from one set of experiments. At least four repetitions are needed. b) A quantitative analysis by image processing is required here.

Reviewer 5 Report

Authors produced and analyzed coatings of PVTF on Ti metal surface, with potential use in bone tissue substitution, in term of chemical and mechanical characterization, in addition biological toxicity was assessed. The manuscript is well written, however some issues should be clarified, as listed below:

1- XPS analysis:

a)      what kind of software was used for peaks deconvolution?

b)      Why they analysed only O region? I suggest to verify if in Ti in order to compare Ti-O contributions to the peak with the Ti-O species found in the O region, also in term of mass balance.  

c)        raw 211-212, authors denoted  TiOHT species with Bridging OH group, and TiOHB groups species with Terminal OH group, please invert TiOHT and TiOHB as used in the refernces article, in to do not confuse the reader.

2- FTIR: in the spectrum several signals are present, please try to assign also the ones not attributable to β-phase

3- AFM: Please explain how authors calculated roughness

Round 2

Reviewer 2 Report

The manuscript has improved a lot overall, but please add this still unfinished, and then it could be accepted:

- You have no indicated the AFM probes used, only the material.

- Since the AFM images are small, you can leave them like this, but since the scale in z is illegible even if it is enlarged, indicate in the figure 5 caption the minimum and maxima values of z corresponding to each panel a, b, c, d, e and f.

- The name of samples in axe “x” of new figure S1 is illegible, please increase its size.

In general it is fine, but can be improved

Reviewer 3 Report

The authors have addressed most of my comments from the previous review

Author Response

Thanks for your suggestion.

Reviewer 4 Report

The corrections are satisfactory.

Author Response

Thanks for your suggestion.

Reviewer 5 Report

Authors did not complete the suggested revisions. Comment are in the attached file
